# Progress on the Industrial Applications of Red Mud with a Focus on China

**Hua Zeng** [1,2]**, Fei Lyu** [1,2]**, Wei Sun** [1,2]**, Hai Zhang** [3,4]**, Li Wang** [1,2,*]  **and Yanxiu Wang** [1,2,*]

1. School of Minerals Processing and Bioengineering, Central South University, Changsha 410083, China; zenghua@csu.edu.cn (H.Z.); lvfei2012@csu.edu.cn (F.L.); sunmenghu@csu.edu.cn (W.S.)
2. Key Laboratory of Hunan Province for Clean and Efficient Utilization of Strategic Calcium-Containing Mineral Resources, Central South University, Changsha 410083, China
3. Guizhou Provincial Geological and Mineral Bureau, Liupanshui 553001, China; zhanghai01504130@163.com
4. Guizhou Dikuang Sanxi Resources Technology Co., Ltd., Liupanshui 553001, China
* Correspondence: li_wang@csu.edu.cn (L.W.); wangyanxiu@csu.edu.cn (Y.W.)

**Abstract:** Red mud (RM), also called bauxite residue, is a strong alkaline industrial waste generated during the alumina production process. The annual production of RM in China is large, but its average utilization rate is low (only 4%). High generation and low consumption make the disposal of RM mainly by stockpiling, which has caused serious heavy metal pollution and radioactive contamination. In this paper, the various industrial utilization methods of RM in China during the past 60 years have been introduced. Moreover, some recent industrial progresses were referred. The results show that RM can be widely used in building materials, valuable metals extraction, and some novel utilization methods, such as silica-calcium fertilizer, inorganic polymer material and desulfurizer. Most of the industrial utilization methods of RM have been used until now and some successfully applied to other aluminum plants, providing some feasible routes for a large amount utilization of RM. Some industrial utilization methods (such as oil well cement and calcium silicon fertilizer) have not been used due to some problems that cannot be ignored, but it provided a lot of valuable experience and was helpful for the subsequent RM utilization. Moreover, some novel and feasible RM utilization methods were proposed and successfully industrialized, which showed that RM has a broader application prospect. Many actual practices showed that the best way to safely dispose of RM was to develop technology that could consume large amounts of RM or transform it into secondary resources, which may need more time and effort.

**Keywords:** red mud; resources recovery; industrial utilization progress; construction materials; Fe extraction

## 1. Introduction

Aluminum, as one of the most abundant metal elements on the earth, is widely used due to its excellent properties and huge bauxite reserves, of which global bauxite resources are estimated to be between 55 and 75 billion tons. Primary aluminum production reached 63.70 million tons in 2019, making it second only to steel in consumption (raw steel production reached 19 billion tons in 2019) [1,2]. Figure 1 shows the change in alumina production from China and other countries or regions in the past 20 years. The world's alumina production has been increasing steadily, with output in 2019 reaching about 132.3 million tons (t), up about 1.3 times that of 2003. Those alumina or aluminum was produced by bauxite ore, and some aluminum could be obtained by recycling. For example, aluminum produced from old scrap in America reached 190 million tons in 2019 [2]. China's alumina output has been growing rapidly in the past two decades, and the output of alumina in 2019 has reached about



71.28 million tons, increased by about 10.7 times compared with that in 2003. China's global share of alumina production also increased from 10.4% in 2003 to 53.9% in 2019, which made China the world's largest alumina producer.

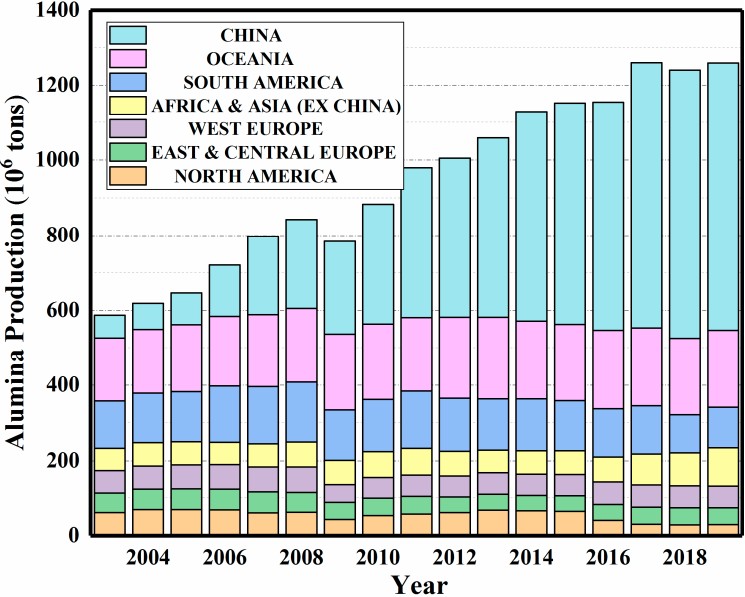

**Figure 1.** Global alumina accumulated amounts during recent twenty years (data from the International Aluminum Institute [3]).

The rapid increase in alumina production followed with the fast-growing production of its by-product, red mud (RM). RM, also named bauxite residue, is the major waste produced during the alumina production process [4]. Depending on the quality of the bauxite and its processing [5], about 15–40% of processed bauxite ore enters the waste in the form of alkaline slurry [6]. It is estimated that 1.0–1.5 t of RM was generated when produced 1 t of alumina [7,8]. The global annual production of RM has already exceeded 160 million tons [9], of which China accounted for about 60% [10]. Unfortunately, high alkalinity, complex compositions and other properties make the use of RM difficult. The average utilization factor of global RM was 15%, and that of China was only 4% [11]. The utilization factor of RM was lower, and RM was even not used at all in some areas. Up to now, global RM reserves have exceeded 4.0 billion tons [12–14], and the cumulative amount of RM in China has also exceeded 350 million tons [15]. Moreover, the annual increase amount of RM in China has exceeded 100 million tons [16,17].

High production and low consumption made the disposal of RM mainly by damming stockpiling (as shown in Figure 2), and the type of RM accumulation could be roughly divided into wet stockpiling, semi-dry stockpiling and dry stockpiling according to the water content of RM slurry [8]. The water content of RM slurry deposited by the wet method was generally higher than 70%, and that of RM slurry deposited by the semi-dry method was usually 40–60%, and that of RM slurry deposited by the dry method was normally required to be lower than 35% [18]. Additionally, the wet method has been gradually replaced by the dry method, which may be due to the high dam body stability and large effective storage capacity of dry stockpiling [19]. Moreover, long-term RM damming stockpiling would occupy huge land areas (approximately 1 km$^2$ per 5 years for a 1 Mtpy alumina plant [20]), which caused the reduction of farming land [21]. Moreover, RM contains many heavy metals (Cd and Cr), radioactive elements (Th, Hf, U, TR) and fluoride. RM easily seeps into ground and underground water, and drifts into air [6], and caused the excessive content of these toxic substances, seriously polluting the surrounding soil, air and groundwater [22]. Moreover, the cost of RM disposal accounted for about 2% of the alumina price [6], which was hard to afford for many aluminum companies and government.

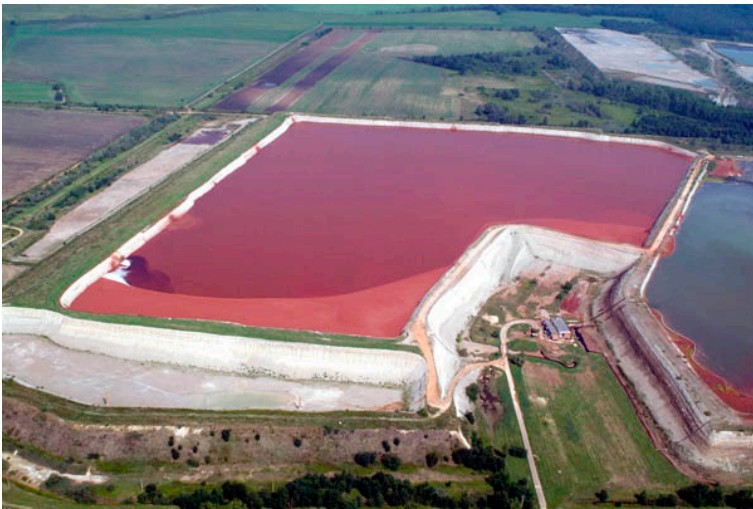

**Figure 2.** Planform of red mud (RM) yard disposed by damming stockpiling [2].

RM is also a potential secondary solid resource due to it containing a large number of valuable components, such as Al, Fe, Ti and Sc [23]. During past several decades, researchers have been working on developing environmentally applicable and cost-effective methods to solve the problems involved in RM utilization. Up to today, RM were mainly studied in many ways, including: (1) manufacturing construction materials, such as cement and concrete [24–26], brick [27–29], ceramic [30–32] and pavement [9,33]; (2) removing heavy metals from wastewater solutions [34,35]; (3) extracting valuable metals from RM, such as Fe [36], Ti [37] and Al [38]; (4) producing various catalysts [39–41].

Many achievements for RM harmless utilization have been obtained during the past several years. However, most of these achievements were only in the laboratory stage, which may be due to there being some non-negligible problems. Problems such as a high cost of raw material, safety and environmental issues, low public acceptance and limited market made only a small number of researches successfully achieved industrialization (such as cement and subgrade [42,43]). This paper focused on the industrial utilization methods of RM in different regions in the past 60 years, which are shown in Figure 3. Moreover, some recent industrial progresses were also introduced.

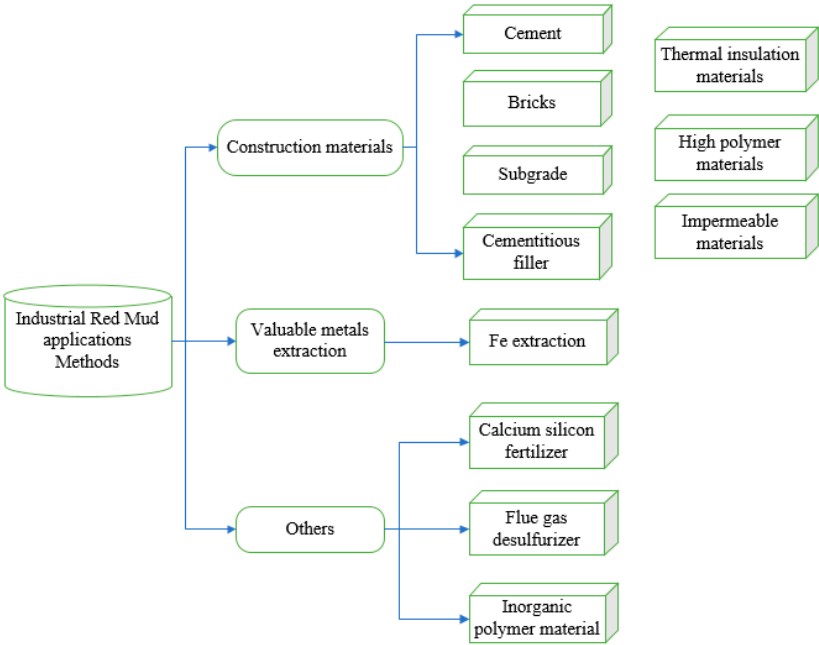

**Figure 3.** Industrial RM applications methods in different fields during past 60 years in China.

## 2. Characteristics of RM

It is important to analyze the mineral compositions and physico-chemical properties of RM as its applications are based on those characteristics [9]. Moreover, the characteristics of RM are closely related to the compositions of bauxite ore and its processing methods. The type of bauxite ore could be basically divided into diaspore, boehmite and gibbsite, the latter two of which are the main bauxite types. Both of them are rich in iron and aluminum content and poor in silicon content. Additionally, those compositions made the bauxite good in alkaline solubility [44]. Additionally, it was usually dissolved by the Bayer method and it was estimated that about 95% bauxite ore was processed to produce alumina by the Bayer method [45]. Table 1 showed the main chemical compositions of Bayer process RM in different plants worldwide. It could be seen that Bayer process RM was rich in $Fe_2O_3$, $Na_2O$ and $Al_2O_3$ content and poor in CaO content, which made the use of RM in some areas, such as Fe extraction, promising.

**Table 1.** Main chemical compositions of Bayer process RM in different plants worldwide (%, *w/w*).

| Country | Plant | Main Composition (%) | | | | | | References |
|---|---|---|---|---|---|---|---|---|
| | | $Fe_2O_3$ | $Al_2O_3$ | $TiO_2$ | $SiO_2$ | $Na_2O$ | CaO | |
| China | Shandong | 34.28 | 21.69 | 7.61 | 13.70 | 10.96 | 0.67 | [46] |
| China | Guizhou | 20.74 | 20.73 | 5.29 | 17.19 | 6.39 | 15.85 | [47] |
| China | Henan | 45.6 | 17.6 | 5.8 | 11.5 | 3.7 | 13.2 | [48] |
| Iran | Jajarm Alumina Industry | 19.66 | 17.9 | 4.9 | 14.4 | 7.2 | 14.8 | [49] |
| Greece | Aluminium of Greece | 42.58 | 16.63 | 5.00 | 7.60 | 3.49 | 11.36 | [50] |
| Australia | AWAAK | 28.5 | 24 | 24 | 18.8 | 3.4 | 5.2 | [51] |
| Brazil | Alunorte | 45.6 | 15.1 | 15.1 | 15.6 | 7.5 | 1.2 | [51] |
| Germany | AOSG | 44.8 | 16.2 | 16.2 | 5.4 | 4.0 | 5.2 | [51] |
| Spain | Alcoa | 37.5 | 21.2 | 21.2 | 4.4 | 3.6 | 5.5 | [51] |
| USA | RMC | 35.5 | 18.4 | 18.4 | 8.5 | 6.1 | 7.7 | [51] |
| India | NARCO | 51 | 18 | 9.8 | 4.6 | 5.3 | 1.8 | [9] |

Diaspore is the main bauxite type in China, which is rich in aluminum and silicon content and poor in iron content. This type of bauxite is poor in dissolution performance. Additionally, the aluminum companies in China (except Guangxi Pingguo Aluminum Company, which adopts the Bayer method), all adopt the sintering method or combined method to produce alumina. Table 2 shows the main chemical compositions of RM from the sintering process and combined process in different regions in China. It can be seen that China's sintering and combined RM are rich in $SiO_2$, $TiO_2$ and CaO content when it is poor in $Al_2O_3$, $Fe_2O_3$ and $Na_2O$ content, making RM promising to be used in some areas, such as construction materials. RM also contains an array of minor elements, and typical rare-earth element concentration and radioactive elements of RM are shown in Table 3. Some of these elements were harmful and made the industrial utilization of RM difficult.

**Table 2.** Main chemical compositions of RM from sintering process and combined process in different regions in China (%, *w/w*) [52,53].

| Compositions | Sintering Process | | | | | Combined Process | |
|---|---|---|---|---|---|---|---|
| | Shandong | Guizhou | Shanxi | Henan | Zhongzhou | Zhengzhou | Shanxi |
| $SiO_2$ | 22.00 | 25.90 | 21.43 | 21.28 | 21.36 | 22.50 | 20.63 |
| $TiO_2$ | 3.20 | 4.40 | 2.90 | 3.39 | 2.64 | 7.30 | 2.89 |
| $Al_2O_3$ | 6.40 | 8.50 | 8.22 | 6.96 | 8.76 | 7.00 | 9.20 |
| $Fe_2O_3$ | 9.02 | 5.00 | 8.12 | 12.29 | 8.56 | 8.10 | 8.10 |
| CaO | 41.90 | 38.40 | 46.80 | 39.82 | 36.01 | 44.10 | 45.63 |
| $Na_2O$ | 2.80 | 3.10 | 2.60 | 2.41 | 3.21 | 2.40 | 3.15 |

**Table 3.** Typical concentrations of trace, rare-earth and radioactive elements in RM [23,54,55].

| Composition | Concentration (mg/kg) |
| --- | --- |
| U | 50–60 |
| Ga | 60–80 |
| V | 730 |
| Zr | 1230 |
| Sc | 60–120 |
| Cr | 497 |
| Mn | 85 |
| Y | 60–150 |
| Ni | 31 |
| Zn | 20 |
| Lanthanides | 0.1–1% |
| Th | 20–30 |

Complex and flexible compositions make RM have some special physical characteristics. As the name suggested, RM is a fine-grained substance with different degrees of red. The average particle size of RM is below 10 μm [22] and typical values would account for 90% volume below 75 μm [56], which mainly rely on the grinding degree of bauxite [43]. Different degrees of reddish color of RM come from the large amount of $Fe_2O_3$ [57]. RM is also a highly alkaline substance, and the amount of free alkali reaches about 2–3 g/L (calculated by $Na_2O$), resulting in a pH value between 10 and 13 [58,59]. Due to the high fineness, RM presents a porous surface [50] and high water absorption, and the specific surface area (BET) and water content can reach 64–187 $m^2$/g and 700–1000 kg/$m^3$, respectively [60]. Additionally, it is a light mineral with a specific gravity of 2.6–3.5 g/$cm^3$ [61]. Moreover, the void ratio of RM is 2.5–3.0, with high compressibility (Eg = 28–40 MPa) and low shear strength (C = 9.6–74.3 kPa; φ = 13.5–21.0°) [60].

## 3. Construction Materials

### 3.1. Cement

Ordinary Portland cement is generally sintered from a mixture of limestone, quartz, iron ore, clay or bauxite [42]. RM has some cementitious activity, and its compositions are similar with Portland cement, which means that RM can be used to produce cement. However, its hydraulic activity is low [55], so RM is usually synergistic with gypsum, blast furnace slag, fly ash, coal gangue, lime and other substances to stimulate its cementitious activity.

Shandong Aluminum Company, as China's first alumina plant, began to use RM very early. As early as 1958, the company successfully produced Portland cement using sintering RM and limestone tailings as the main raw materials, adopting a wet production process, and 5% gypsum and no more than 15% active mixed materials were also added [62]. The producing process of RM-based cement was similar with Portland cement, which is shown in Figure 4. The strength of RM-based cement reached 48–52 MPa, fully meeting the requirements of ordinary 425R cement. However, due to the high alkali content of RM (2.0–10.0% $Na_2O$) [63], the addition ratio of RM was restricted and only about 25% [64].

In order to increase the addition ratio of RM in raw material without removing alkali, the company carried out many industrial experiments of adding mineralization agent to cement. The results showed that adding 0.5% fluorite mineralizing agent (containing 40–60% $CaF_2$) to raw materials can promote the normal absorption of free calcium oxide in the sintering process. The clinker could be sintered normally with high alkali content (up to 1.8%), and the proportion of RM can be increased to 40–45% without scarifying the mechanical properties [65]. This technology was successfully applied to production in the early 1980s.

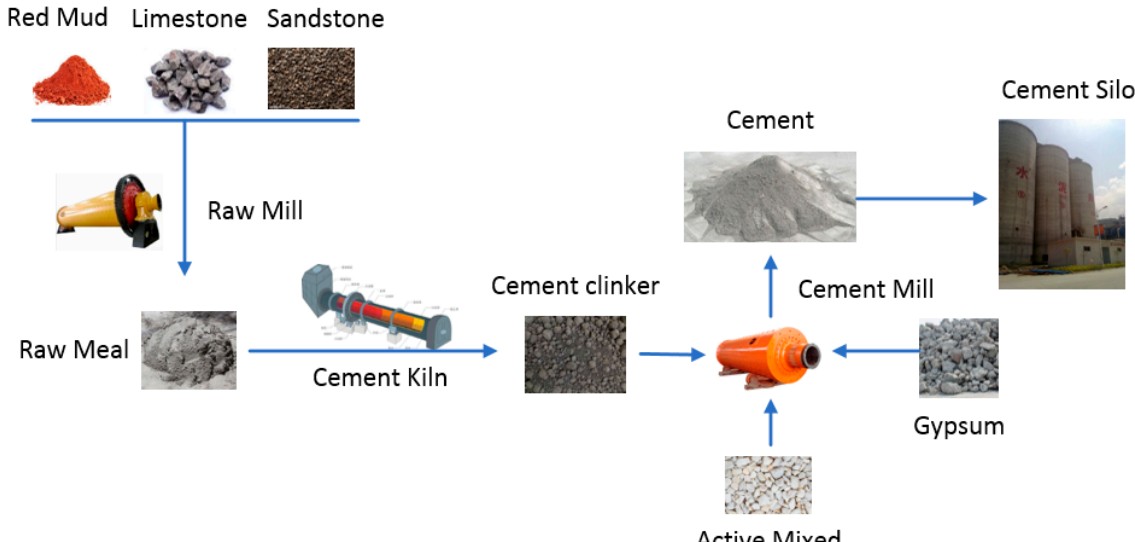

**Figure 4.** Preparation process of RM-based Portland cement.

Since 1954, Shandong Aluminum Company has been producing RM-based Portland cement with an annual output of 1.6 million tons of clinker. By 1997, the company had produced more than 20 million tons of cement and comprehensively utilized more than 6 million tons of RM, which is by far the largest RM comprehensive utilization method. The annual average ratio of RM was 20–38%, the utilization amount of RM was 200–420 kg/t, and the comprehensive utilization rate of RM was 30–55% [66].

However, the wet process was to grind raw materials into slurry (water content was 32–40%) for calcination, requiring a lot of energy to dry raw material and making its cost high [67]. Dry process is to grind raw materials into dry powder for calcination with low energy consumption. However, it is difficult for the dry process to make raw materials fully mix, limiting the application of the dry process. With the technological progress in the cement industry, it is possible to replace the wet process with the dry process. Moreover, Shandong Aluminum Company solved the problem of RM alkali removal. They used lime as main dealkalization agent to carry out dealkalization at a low temperature of 70–90 °C, a calcium/sodium ratio of about 3.0 and a liquid/solid ratio of 3.0. The dealkalization rate of RM reached more than 80%, and RM alkali was successfully reduced from 2.5–3.5% to less than 1.0%, satisfying the requirement of low alkali cement [68]. In 2003, the company's first 3000 t/d new dry cement clinker production line was put into operation using sintering RM and combined RM as raw material. The addition ratio of RM was up to 45%, and the quality of cement was improved from number 425 Portland cement to number 525 Portland cement [69]. Drawing on a similar process, Guangxi Aluminum Company started to build a new dry cement production line of 3200 t/d clinker and 6600 t/d clinker in 2015. It was estimated that the company would discharge 1.02 million tons of RM, limestone waste residue, fly ash, slag, desulfurization gypsum and other industrial waste residue every year after the completion of the project.

Apart producing Portland cement, RM was also used to produce other cement. Taking advantage of the strong resistance to sulfate erosion, Shandong Aluminum Company developed sulfate-resistant RM cement and the consumption of RM of cement reached 600–800 kg/t. The products were mainly used for salinization projects with seawater contact, anti-corrosion and underwater engineering of the salinization industry. The addition ratio of RM was up to 60%, and the output reached 2000–4000 t/year [65]. On the basis of using RM to produce Portland cement, the plant adjusted the grinding size of cement powder and properly controlling the content of tricalcium aluminate in clinker, and successfully produced oil well cement. The production capacity reached 100,000–200,000 t/year and produced oil well cement was mainly used in the SINOPEC Shengli oilfield [65]. Entering the

1990s, the oil well cement of Shandong aluminum plant was suspended due to the Shengli oilfield building its own cement factory [65].

Long–time industrial practices showed that RM can be used to produce Portland cement and other cements. Produced RM Portland cement also showed comparable mechanical properties compared with ordinary Portland cement. Moreover, RM Portland cement has some better characteristics, such as high early strength, moderate setting time, good freezing–thawing resisting performance, good sulfate resistance and erosion resistance. It is suitable for components requiring high construction speed and high early strength and a working environment requiring high sulfate content.

The high alkalinity of RM is the main problem limiting its utilization in the cement industry (even building industry). The alkali in RM mainly exists in the state of soluble alkali and chemical binding alkali. Soluble alkali is an important component of RM alkalinity, among which soluble alkali accounts for about 40–60% of the total alkali amount. The soluble alkali mainly exists in the forms of $NaOH$, $Na_2CO_3$, $NaHCO_3$, $NaAl(OH)_4$, $Na_2SiO_3$, $KOH$ and $K_2CO_3$ [70]. Additionally, it is easily soluble in the liquid phase, leading to the increase in pH value. The chemical binding alkali is the sodium silicon slag, which was formed by the reaction of $SiO_2$ in bauxite with sodium aluminate solution. Additionally, chemical binding alkali mainly existed in the forms of cancrinite, sodalite, zeolite and amorphous sodium aluminosilicate [71]. The cement industry is strict with the alkali content of raw materials, which is generally required to be less than 1%. It was showed that the addition ratio of RM was mainly depended on the amount of alkali substances existed in RM (only about 20–30%). Although some feasible de-alkali method was putted, the high alkalinity of RM is still the main problem limiting the use of RM in the cement industry (even whole building industry). Moreover, it could be found that the market acceptance of RM products is low from the suspension of oil well cement. So, in addition to making RM cement have a higher performance, it was also necessary for government to increase the publicity of RM products and encourage the use of RM products.

### 3.2. Sintered Bricks

Traditionally, sintered bricks were made from clay, shale, coal gangue and other substances fired at a high temperature [72]. The production process of sintered RM brick was similar to the traditional sintered brick, which were normally produced using RM, shale and other materials as main materials after comminuting, mixing, shaping, drying and roasting. RM has been successfully used to produce various bricks such as insulation brick, permeable brick, landscape brick and pavement brick.

The compositions of RM are relatively stable and can be regarded as an inert component when temperature is less than 900 °C [30], which makes the use of RM as raw material to produce thermal insulation and refractory insulated material possible. Long time studies showed that RM brick has good properties and its radioactivity was in an acceptable range (internal exposure index ($I_{Ra}$) and external exposure index ($I_r$) was less than 1.0 according to GB 6566-2010 (Limits of radionuclides in building materials)) [2].

In 2009, Shanxi aluminum plant successfully developed refractory thermal insulation bricks for industrial kilns by using RM and fly ash as raw material, and the addition ratio of RM and fly ash was more than 50%. A production line of RM fly ash refractory thermal insulation bricks with a production capacity of 12,000 t/year was built in 2010 [73]. Up to now, the company has formed a production line of RM fly ash refractory insulation bricks with an annual output of 100,000 t, and the addition ratio of RM accounted for about 30% of the quality [73]. The annual utilization amount of RM reached about 30,000 t, and the RM fly ash firebrick showed comparable mechanical properties compared with traditional sintered brick. Moreover, it has many advantages, such as high porosity, low density and low thermal conductivity.

For expanding the application range of sintered RM bricks, Guizhou Building Materials Science Research and Design Institute successfully produced sintered RM pavement brick using RM, shale and coal gangue as the main raw materials, among which the mixing amount of RM was up to 40–50%, and the mechanical properties satisfied the requirement of sintered pavement brick (GB/T 26001-2010).

The average compressive strength was more than 25 MPa and frost would not occur. Moreover, the radioactive elements were satisfied with the requirement of limits of radionuclides in building materials (GB 6566-2010). At present, an industrial production line, with an annual output of 40 million RM sintered pavement bricks, has been built. Apart from using sintering RM to produce sintered bricks, a company in China's Shandong province also successfully made use of Bayer RM to produce landscape bricks in 2018 by adding a small amount of ceramic raw materials, among which the addition amount of RM was more than 50%. The average compressive strength reached about 80 MPa, and the resistance of acid and alkaline was over 90%. The internal exposure index ($I_{Ra}$) and external index (Ir) were 0.2 and 0.9, respectively, both of which met the requirement according to the limits of radionuclides in building materials (GB 6566-2010) (Ira and Ir was required to be less than 1.0). At present, the company is docking with the local aluminum plant for the use of this achievement.

RM permeable brick belongs to another patent technology of RM sintered brick [74]. The technology solves the problem of the dissolution of alkali metals and heavy metals in RM, so that RM can be safely used as raw material in permeable pavement materials, and the comprehensive utilization rate of RM waste residue can reach 50–70%. In 2018, a company in Shandong province built a fully automatic production line of ecological RM permeable brick with a daily output of 3000 square meters, which can digest nearly 100,000 t of RM every year, and the addition ratio of RM reached 80%. The thickness of RM permeable brick is 3.5 cm, and the bending strength of RM brick reached 6–8 MPa when the national standard is 3.2 MPa. Radioactivity, heavy metal dissolution and other indicators met the requirements of building materials according to GB 6566-2010. Moreover, RM brick has excellent performance in terms of water permeability, anti-freezing and thawing, and anti-sliding property [42]. Economically, the price of RM brick was the same as that of clay brick. Until now, the brick has been widely used in pavement laying (Figure 5), greatly improving pavement permeability, anti-sliding and noise reduction.

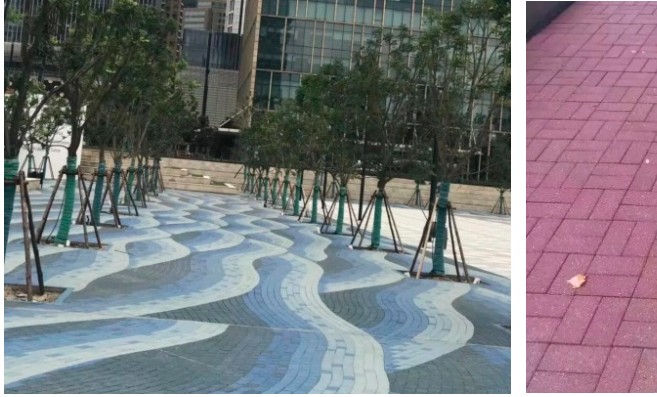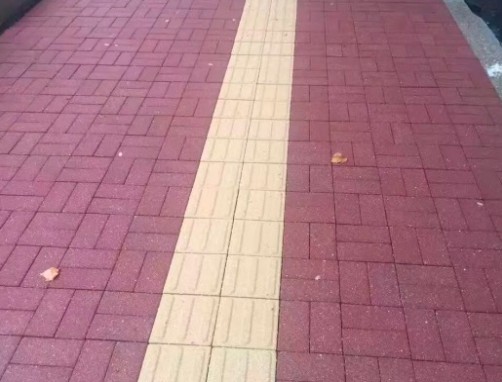

**Figure 5.** RM permeable brick lay in somewhere in Shandong province.

### 3.3. Non-Fired Bricks

In the calcination process of sintered bricks, a large number of environmental pollutants were released, large amounts of energy consumption were consumed, and many greenhouse gases were produced [19]. Nowadays, the production of sintered bricks has been restricted, even have forbidden producing clay bricks in global some areas, which made the production of non-fired bricks become the development direction in the brick industry. RM contains a large amount of dicalcium silicate ($C_2S$), some tricalcium silicate ($C_3S$) and tricalcium aluminate ($C_3A$) phase, which makes RM have some potential cementitious activity and water hardness [74]. The active index of RM is more than 90%, and it can be used to produce non-fired bricks by adding some activators, curing agents and aggregates. Gel materials are generated through a series of hydration reactions and make the combination between different particles closer, so as to meet relevant national standard of baking-free brick [75].

Shandong Aluminum Company successfully produced RM non-fired brick by natural curing and autoclave curing using sintering RM, fly ash and mining slag as the main materials. The raw materials of the two processes have been shown in Table 4. The 28-day compressive and flexural strength of non-fired brick by natural curing and autoclave curing reached 17.5 and 2.8 MPa, 15.6 and 4.8 MPa, respectively, meeting the requirement of MU15 brick according to relevant Chinese standards (GB 11945-1999). Both types of non-fired brick are already in production. Economically, the production cost with natural curing could be controlled below 0.11 Yuan/block, and the production cost with autoclave curing could be controlled below 0.14 Yuan/block, which showed better economy [76].

**Table 4.** Raw materials of bricks by natural curing and autoclave curing [77].

| Process | Raw Material (wt %) | | | | | | |
|---------|--------|--------|---------|------|------|--------|--------|
|         | **Wet RM** | **Dry RM** | **Fly Ash** | **Sand** | **Lime** | **Gypsum** | **Cement** |
| Natural curing | 21.7 | 9.3 | 24 | 2 | 10 | 5 | 1 |
| Autoclave curing | 21 | 8 | 25 | 29 | 10 | 5 | 1 |

The company also successfully developed a non-steaming and non-firing brick based on RM:fly ash:aggregate of 30%:23%–30%:30%. The 28-day compressive strength was more than 9 MPa, which was not satisfied with the requirement of MU15 bricks [78]. Then they took 50% fly ash, 10% calcium carbide slag, 10% sodium silicate slag, 20% RM, 10% stone debris and 2% desulfurization gypsum as raw materials, and successfully made RM brick using the semi-dry pressing molding process. The 28-day compressive and bending strength was up to 16.6 MPa and 4.2 MPa [79], meeting requirement of MU15 brick. In cooperation with Huazhong University of Science and Technology, the company took RM and fly ash as the main raw materials, and prepared RM fly ash brick adopting natural curing. Moreover, some lime, gypsum and 32.5 ordinary Portland cement were added as curing agent and activator. The ratio of RM:fly ash:aggregate:gypsum:lime:cement are 33%:24%:30%:2%:10%:1%. The bending strength and compressive strength reached 3.22 and 22.42 MPa, meeting the requirement of MU15 brick according to GB 11945-1999. The brick has been already in production, 18 million of RM fly ash bricks were produced every year and more than 20,000 t of RM was consumed annually [80].

Many actual practices showed that RM can be used to produce sintered bricks and non-sintered bricks. Additionally, taking advantage of the high porosity of RM, RM bricks are more suitable for porous bricks. Although there are some feasible RM utilization methods in the brick industry, there still are some key problems that need to be solved. Based on the above, it could be found that the RM addition ratio was low (only about 20–30%), and it was hard to take advantage of the high waste consumption of brick. Moreover, the high alkalinity of RM still poses a big problem for its utilization in building materials, which easily caused efflorescence. Additionally, the high cost of activator and the radioactivity of RM brick still limited its utilization. How to coordinate the use of RM and other industrial wastes to the greatest extent to make bricks and realize the complete control of waste by waste may be the future development trend.

### 3.4. Subgrade

Not only can the application of the piled RM to the roadbed filling achieve the large-scale utilization of RM, but also it can save the natural resources [8]. However, some properties of RM, such as its high fineness, strong water holding capacity and poor water stability, make the engineering properties of RM cannot meet the technical requirements of subgrade filling. RM is needed to be improved mainly by chemical method. The chemically improved method is normally to change the microstructure of soil by adding lime, cement, fly ash and other inorganic cementitious materials, which is to react physically and chemically with soil particles and then improve the strength, stiffness and water stability of RM subgrade [43].

Shandong Aluminum Company built a 4-km-long RM roadbed demonstration section in 2015 (as seen in Figure 6), which used sintering RM, fly ash and lime as the main raw materials and the ratio was 75%:15%:10% (dry mass ratio). The 7 and 28 day average compressive strength reached 1.2 and 3 MPa, which met the strength requirements of grade-I lime-stabilized soil and highway [81]. This was the first sintering RM pavement base project applied in the actual highway in China, which consumed more than 20,000 t of sintering RM. It was one of the projects with the largest utilization amounts of RM in recent years and has been in normal use until now [82]. In 2008, the company provided the same formula and cooperated with local municipal highway bureau to build a road with 500 m long and 27 m wide, consuming nearly 4000 t of sintering RM. Actual practices showed that the whole test section was in good operation and basically qualified with national standards. [83]. The 7 and 28 day compressive strength reached 1.96 and 2.6 MPa, which met the strength requirements of the first grade of the stable soil layer of lime and the expressway. Economically, the construction cost of RM base was 10–20 Yuan cheaper than several common base materials, such as limestone soil and lime-ash gravel, under the same transportation distance [84].

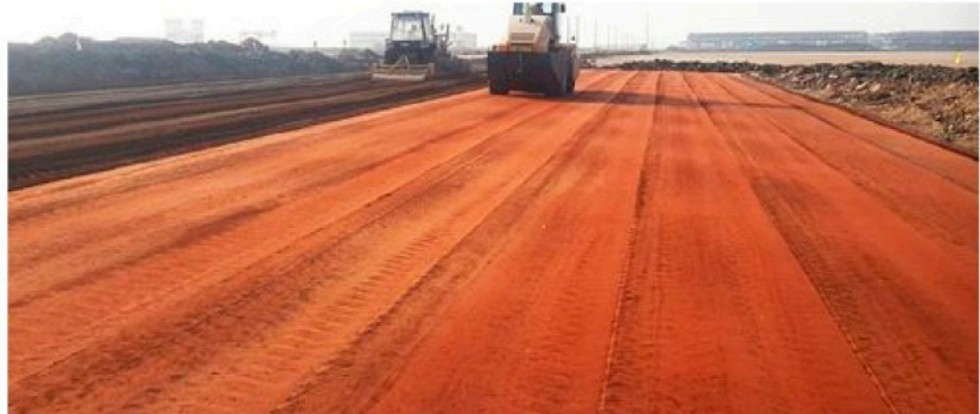

**Figure 6.** Laying site of RM roadbed lay in somewhere in Shandong province.

Cooperating with Beijing Mining and Metallurgy Research Institute, Pingguo Aluminum Company has also developed the first RM basic road and a new RM concrete pavement in China through the comprehensive solidificaion technologies, including alkali stabilization, ion exchange, RM activation and pressure molding. The industrial test of 800 m RM basic road, 500 m RM concrete pavement and 5 km expansion industrial test have been completed. The main raw materials of basic road were RM:lime:fly ash with the ratio of 80%:10%:10% and the curing agent were provided by Beijing General Research Institute of Mining Industry. The density was 1.84 g/cm$^3$, and the 7-day and 28-day compressive strength reached 3.55 MPa and 4.25 MPa. The ratio of concrete pavement was RM:fly ash:cement:graded gravel:curing agent = 30–35%:5–10%:10–15%:40–50%:0.2–0.25% [85]. RM was consumed at 2.04 t/m for the basic road and 0.63 t/m for the concrete pavement, with a total RM consumption of 1947 t [86]. After nearly a year test like sun exposure, rain erosion and large-tonnage vehicles unbalanced driving, the operation was excellent, meeting the requirements of high-grade highway. In recent years, Pingguo Aluminum Company has been promoting this achievement, and has been laying 22 km RM roads and totally consuming 117,500 t of RM [86].

A company in China's Shandong province has realized the world's first application of Bayer RM in the construction of actual highway engineering, and successfully applied it to the expressway. In 2018, 25,000 t of RM was used on the subgrade of the expressway and the construction diagram of RM pavement has shown in Figure 7. The RM based concrete was built through mixing and compacting, which was based on RM and RM modifier in a ratio of 92:8. RM modifier was made based on phosphorus gypsum and cement as main material, also some heavy metal reducing agent, complexing agent and curing modification materials were added. The RM modifier had a good solidification and stability effect on the pollutants in the original RM, and effectively reduced the dissolution of pollutants,

making RM concrete meet the requirements of environmental protection. The 7 d compressive strength reached 3.1 MPa, fully meeting the requirement of the highway. The subgrade has been successfully applied to the highway, and the company's achievements have been successfully used in many RM highway projects, expansion projects and the national highway G309 construction project in China. Modified RM could be used to replace ordinary lime soil and cement, and practical utilization showed that the price was 10–30% cheaper [87].

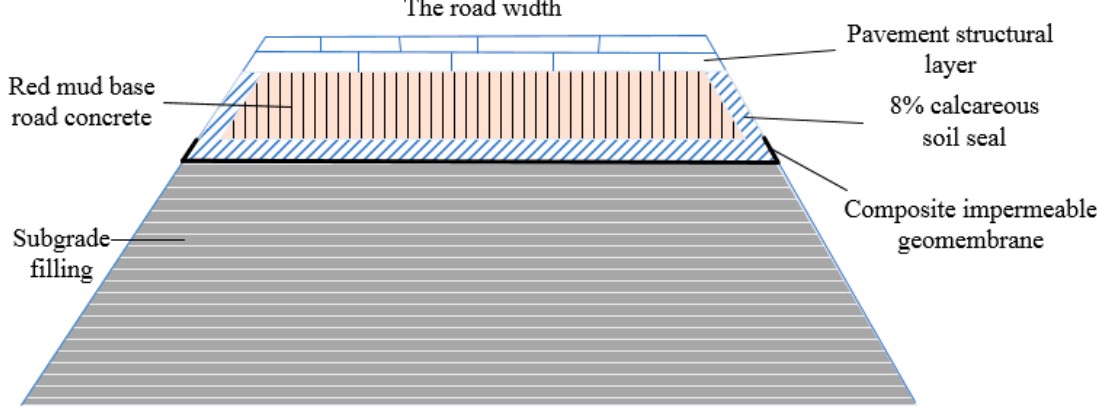

**Figure 7.** Construction diagram of RM pavement [87].

The University of Science and Technology, Beijing, successfully produced a new pavement material using RM, coal gangue, fly ash, desulfurization gypsum, furnace slag and a small amount of cement. The addition ratio of solid wastes is as high as 97%, and the mechanical properties were good. The 7-day compressive strength was up to 6.2 MPa, the leaching of sodium element in 7-day was 0.0013% (mass percentage), and the construction process was convenient. This achievement has been successfully applied in the road project of Huaxing Aluminum Company in China's Shanxi province, and actual practices showed that its application effect was good.

With the characteristics of small density, fine particles and good hydration activity of sintering RM, Shandong Transportation Research Institute prepared high fluidity and light-weight cementitious pouring materials combining sintering RM with some cement, fly ash and water-reducing agent. The ratio of sintering RM:cement:fly ash (mass ratio) was 84:8:8 and the content of water-reducing agent was 1% of the quality of dry powder. The engineering was successfully applied in backfilling on highway expansion project. Long-time actual properties tests showed that the strength and working performance of the pouring entity were good, which has high practicability and application value [88].

A road base course of 50 m in length, 15 m in width and 24 cm in thickness was laid by Henan Institute of Traffic Science with lime:fly ash:RM as 6:24:70. The construction practice and application process showed that lime fly-ash-stabilized RM base had good dynamic stability, dry shrinkage and warm shrinkage [89]. They also used lime:RM:soil as 6:36:58 to lay two lime RM stabilized soil bases with a length of 150 m, a width of 15 m, a thickness of 24 cm and a length of 2000 m and a width of 7 m. A total of 14,000 m$^2$ of RM was consumed [89]. Two years of testing showed that the RM base has the advantages of high strength, good plate property, water stability and frost resistance. Actual practice showed that the cost of lime RM mixture base was about 30% lower than that of lime clay base due to the test section being close to the RM producing area. Moreover, an appropriate amount of RM was also applied to acid soil to improve the acid soil by virtue of the strong alkalinity of RM. This method has been applied in some acid mines [66].

Although some properties make RM cannot meet the requirement of subgrade, but many actual practices of RM in subgrade showed that it can be immensely activated by inorganic cementitious activators such as lime, fly ash, coal gangue and slag. These activators can react physically and chemically with RM, changing its microstructure and improving its strength, stiffness and water stability. At present, RM can be well solidified and would not pollute the environment. However,

in the longer run, it still need a lot of time to confirm. Also it was showed that there are some problems limited the use of RM in subgrade such as the cost and stimulating effect of RM activators, the cost and effect of impermeable material and the solidified ability of RM modifiers.

### 3.5. Thermal Insulation Material

Micro-porous calcium silicate thermal insulation material is a new type of environmental protection and energy saving material. It has some advantages, such as low bulk density and thermal conductivity, high compressive and flexural strength and reusability, and is generally used at high temperature parts of 650–1000 °C [90]. It is normally produced by a dynamic method using materials rich in active silica as main materials and lime, fiber as reinforcement materials [91].

Using RM to replace diatomite and combining it with lime, bentonite and admixture, Shandong aluminum plant successfully developed RM micro-porous calcium silicate insulation material. The addition ratio of RM reached 30%, and the production process and physical properties of the product are shown in Figure 8 and Table 5. Its mechanical properties were excellent (the density was 215 kg/cm$^3$, the flexural and compressive strength reached 0.31 and 0.91 MPa, the thermal conductivity was 0.0554 W/(m·K), and the maxing affordable temperature was up to 650 °C), fully meeting the relevant national standards [91]. The product was put into industrial production in 2001, and the production line of calcium silicate insulation material with an annual output of 12,000 m$^3$ was built. Practical practices showed that the production line process stability, product quality and economic benefits were good [91].

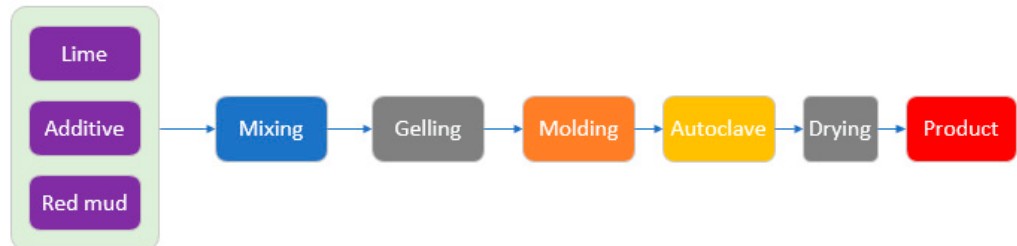

**Figure 8.** Production process of RM micro-porous calcium silicate thermal insulation material [91].

**Table 5.** Properties of RM micro-porous calcium silicate thermal insulation material [91].

| Property | RM Micro-Porous Calcium Silicate Insulation Material | GB/T 10699—1998 | American Standard | England Standard |
|---|---|---|---|---|
| Density/kg·cm$^{-3}$ | 215 | ≤220 | ≤240 | ≤240 |
| Flexural strength/MPa | 0.31 | >0.30 | ≥0.31 | 0.25 |
| Compressive Strength/MPa | 0.91 | >0.5 | ≥0.414 | ≥0.5 |
| Maxing service temperature/°C | 650 | 650 | 649 | 650 |
| Thermal conductively/W·(m·K)$^{-1}$ | 0.0554 (374.6 K) | ≤0.062 (373 K) | ≤0.065 | ≤0.061 (373 K) |
| Line shrinkage/% | 1.5 | ≤2.0 | ≤2.0 | ≤2.0 |
| Moisture content/% | 4.7 | ≤7.5 | | ≤7.5 |

Inorganic fiber is another kind of thermal insulation material made from minerals, and it is generally produced by the ore and coke in suitable proportion through high temperature melting and centrifugation. It is usually sprayed onto the surface of objects as an insulating material. Acidity coefficient (MK) and viscosity coefficient (MB) are commonly used to represent and control its performance [43]. A high MK value has a good thermal insulation effect, and is generally 1.4–1.6 and 1.1–1.4 for rock wool and mineral wool. The larger the MB value, the more viscous the melt, and the fiber is not easy to slim. The MB value is generally between 1.0 and 2.0 [92].

$$MK = \frac{CaO + MgO}{SiO_2 + Al_2O_3} \tag{1}$$

$$MB = \frac{mSiO_2 + 2mAl_2O_3}{mFe_2O_3 + mFeO + mCaO + mMgO + mK_2O + mNa_2O} \tag{2}$$

where *m* is the number of holes.

The chemical compositions of inorganic fiber belong to $SiO_2$-$Al_2O_3$-CaO-MgO system, and RM contains oxides required by inorganic fiber, which could be used to produce inorganic fiber by adding fly ash, carbide slag, blast furnace slag and other solid wastes [92]. A company in Henan province has successfully produced qualified inorganic fiber by mixing RM, fly ash and calcium carbide slag on an industrial scale, and the ratio of RM, fly ash and carbide slag was 35–40%:23–28%:32–37%. The flow chart of the industrial preparation method and the chemical composition of RM inorganic fiber were shown in Figure 9 and Table 6. The thermal resistance of fiberboard with a density of 70 kg/m$^3$ was 2.21 (m$^2$·K·w$^{-1}$), and the fiber formation rate reached about 80% [92].

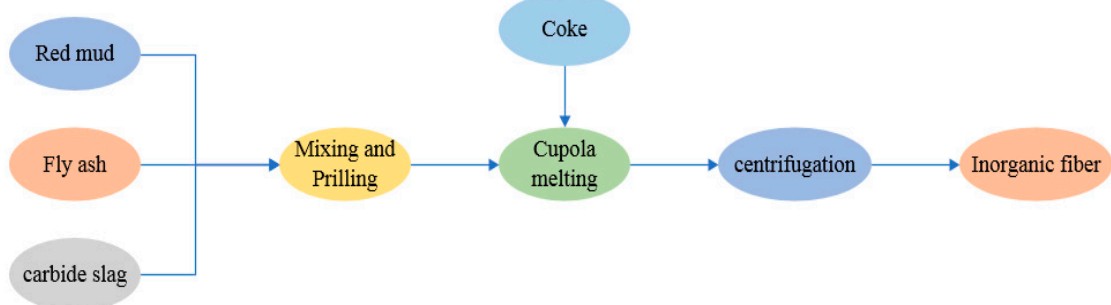

**Figure 9.** Production process of RM inorganic fiber [92].

**Table 6.** Compositions of RM inorganic fiber [92].

| Compositions/% | | | | | | | | MK | MB |
|---|---|---|---|---|---|---|---|---|---|
| SiO$_2$ | Al$_2$O$_3$ | Fe$_2$O$_3$ | TiO$_2$ | CaO | MgO | K$_2$O | Na$_2$O | | |
| 25–35 | 12–21 | 3–9 | 1–2 | 28–35 | 1–7 | 0.5–1.2 | 0.5–2 | 1.1–1.4 | 1.0–1.6 |

A company in Shandong province also successfully produced inorganic fiber using RM as the main material. The RM is mixed with curing agent and acidity coefficient regulator and then fused with coke at a high temperature, and inorganic fiber was obtained after centrifugation. The addition ratio of RM reached more than 70%. Additionally, the pollution of RM was effectively controlled, and the inorganic fiber produced reached the level of rock wool. Economically, the production process was simple and cost low. In 2014, another company in Shandong province successfully produced qualified rock wool with RM as the main raw material and realized industrial production. By adding some additives, the acid coefficient (SiO$_2$, Al$_2$O$_3$):(CaO, MgO) was greater than 1.6, and CaO:MgO was greater than 1. The rock wool has good water resistance, which overcomes the defect of declining stability of slag cotton fiber in wet environment [93].

The high fineness and high BET made RM could be used as thermal insulation material. However, its compositions could not better satisfy with insulation material. It was necessary to add some substances to adjust chemical compositions, which leads to the reduction of RM content. For controlling heavy metal and radioactive elements, some expensive modifier may be needed. It may increase the cost of RM products and make the use of RM uneconomic, making it hard to use on an industrial scale. So, there are still many areas for improvement, such as developing cheaper and/or effective modifier.

## 3.6. High Polymer Material

RM plastic is a new energy-saving and environmentally friendly building material, which provides a new way for the utilization of RM. It was produced based on PVC resin (or waste PVC plastic) as a basis material and pretreated RM as a filling agent after kneading, banburying, calendering or blow molding. Moreover, some waste engine oil and phthalate esters were added as processing aids and plasticizer, and sometimes some glass fiber, plant fiber and synthetic fiber was added as reinforcing

agent [72]. RM plastic normally consisted of 20–80% recycled PVC, 5–80% RM, 0–20% waste oil, 0–50% DOP and 0–20% other fillers [94].

Compared with ordinary PVC plastics, it has higher tensile strength and elastic retention force, higher wear, corrosion, acid and alkali resistance, self-extinguishing ability, better light shielding and aging resistance. Economically, its cost is low, which is 10–20% cheaper compared with general PVC plastic [95]. The mechanism is as follows: (1) CaO, SiO$_2$ and TiO$_2$ in RM are high quality fillers for PVC; (2) RM contains a large number of free alkali and CaO, and can quickly absorb the HCl released from PVC aging and delay the chain reaction. The life of RM plastic is 2 to 3 times longer than that of ordinary PVC products; (3) Fe$_2$O$_3$ and TiO$_2$ in RM are good light shielding agents, which can absorb ultraviolet rays and delay photoaging; (4) RM is combined with waste PVC to form a mesh body to strengthen the plastic, which is a filler with a reinforcing effect on PVC [94]; (5) the fluidity of RM is better than other fillers, so that the plastic has good processing performance [68].

RM plastic has been made into hard and soft products and is widely used in industry, agriculture and daily life (some was shown in Figure 10). Soft RM plastic can be used as storage bags for grain, fruits and vegetables feed [96], and solar water heater [97]. The hard products can be used as corrugated board [98], floor tiles [95], plastic round pipes [95], biogas fermentation tanks and gas storage tanks [99], artificial leather [100], and pipes [101].

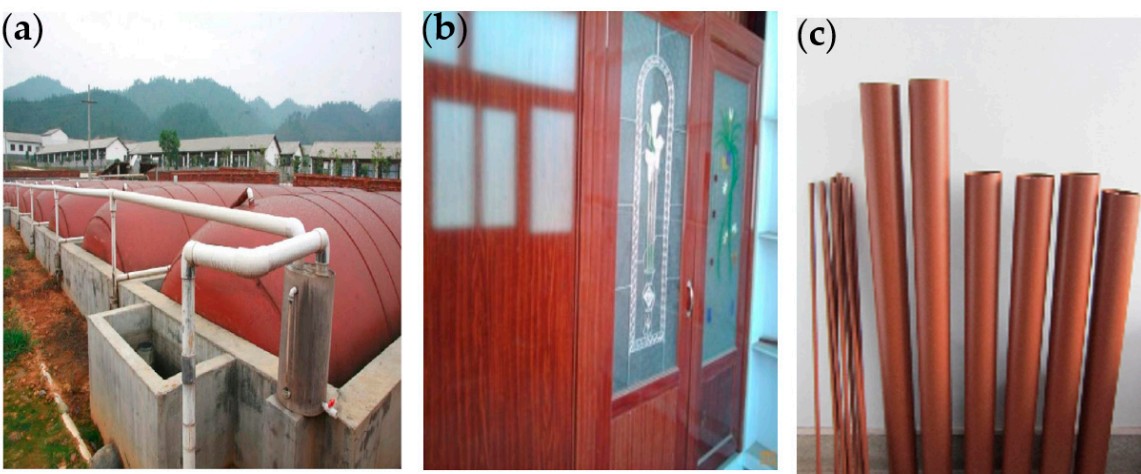

**Figure 10.** Various RM plastic used in different areas ((**a**) biogas fermentation tank; (**b**) plank; (**c**) pipe).

*3.7. Cementitious Filler*

RM contains a large number of hydraulic cementitious minerals, such as dicalcium silicate and tetralcium ferroaluminate, which makes RM have a strong potential activity. However, the particles of dicalcium silicate are covered by the generated calcium silicate hydrate (C–S–H), which greatly reduces the reaction speed of dicalcium silicate. The infiltration and diffusion of reaction molecules are very slow, so that the RM becomes inert, and it only has weak or no hydraulic activity and it cannot be solidified by itself. Some activators such as limestone and gypsum have a significant excitation effect on RM, making RM generate a large number of hydraulic silicates and aluminate gel, and generated sufficient strength [19,102,103].

Shandong Aluminum Industry produced cementitious filling material using sintering RM, fly ash and lime as main materials, in which lime was used as RM activator and the ratio of water:RM:fly ash:lime was 2.43:2:1:0.57. The achievement was successfully applied to the mine filling of its own bauxite mine in the 1990s. Actual practices showed that its application effect is good, and the RM filling material has good working behavior and good fluidity [104]. The setting and hardening speed is faster than cement slurry, and the 60 day average strength of the filled body reached 3.24 MPa [105]. Moreover, the cost was low, only about 20% of the same strength of cement filling materials, which made the RM filling material have a broad application prospect [106].

The use of RM as cementitious filler could achieve high content utilization of RM, and produced filler could reach high strength and high flowability. However, it must consider heavy metal and radioactive elements in RM when use RM in cementitious filler. If these elements could not be solidified, serious second environmental pollution would occur. So, cheaper and/or effective solidifier was needed, and long-term environmental detection was also needed.

### 3.8. Impermeable Material

RM contains a large amount of dicalcium silicate, which makes it a good natural impermeable material. The initial permeability coefficient of RM can reach $1 \times 10^{-3}$–$5 \times 10^{-4}$ cm/s and its impermeability increases with the accumulation time and height of RM in the storage yard. For example, the permeability coefficient of RM can reach $10 \times 10^{-6}$ cm/s at the height of 40 m [107].

Adding lime as activator in RM can accelerate the curing speed and greatly improve the impermeability of RM. At the bottom of its second RM yard, Shandong Aluminum Company used RM and lime as 9:1 and has laid an impervious barrier with a thickness of 0.6–1 m, and its permeability coefficient was $1.55 \times 10^{-7}$ cm/s [108]. For areas with complex geological conditions and high impermeability requirements, a layer of artificial impermeability material can be added to make composite impermeability material [107]. The permeability coefficient of RM can reach $6.32 \times 10^{-9}$ cm/s after adding 8% lime solidified RM and a composite impermeable material formed by artificial impermeable film, which met the impermeable design requirements of RM yard ($<1.0 \times 10^{-8}$ cm/s). This achievement has been successfully applied to the anti-seepage project of RM yard in some aluminum plants [109].

## 4. Fe Extraction

RM contains a large number of valuable metals, such as iron, titanium, scandium and aluminum. Many studies have been conducted on the recovery of valuable metals in RM during the past several years. However, except for the recovery of iron, the recovery of other metals is still in the laboratory stage or preliminary exploration stage. This part will briefly introduce the recycling of iron from RM (especially Bayer RM), which can be basically divided into direct magnetic separation and the reduction of magnetic separation.

### 4.1. Direct Magnetic Separation

Whether RM is generated from imported bauxite or bauxite in Guangxi province, it contains a large number of $Fe_2O_3$ and is mainly in the form of hematite [54]. Hematite is weakly magnetic and can be directly enriched and recovered through strong magnetic separation, which can create some value and reduce the amount of discharged RM.

Aiming at Bayer RM from Indonesia bauxite, Shandong Aluminum Company successfully developed Fe extraction technology using cyclones and pulse high-gradient magnetic machines in 2008. The company divided Bayer RM into two types, one was RM with low Si and Al, and high Fe, and another was RM with high Si, Al and Fe. For different types of RM, different ways of the Fe extraction method were carried out, which has been shown in Figure 11. The company adopted a desands process for the former, the content of $Fe_2O_3$ in RM sand was enriched to about 68.33% by separating 58% tailing using cyclone. The company adopted the process of selecting iron first and desanding later for the latter. The content of $Fe_2O_3$ in the iron powder was 65–75% with an output rate of about 40%, and the iron powder could be used as raw material in the steel industry. While producing iron powder, high iron RM with the content of $Fe_2O_3$ up to 55% was also produced, which can be used as raw material for cement production and an ideal ore for iron powder [110]. The chemical compositions of iron powder and high iron RM are shown in Table 7. In 2009, a production line with an annual consumption of 400,000 t was built, which could produce 80,000 t of iron powder per year. The discharge of RM has been greatly reduced with a reduction rate of 40%, which can reduce the RM storage and maintenance cost of 6.5 million Yuan per year.

**Table 7.** Compositions of iron powder and high iron RM from Bayer RM.

| Compositions | Iron Powder (%) | High Iron RM (%) |
| --- | --- | --- |
| $SiO_2$ | 5.00 | 5.50 |
| $Fe_2O_3$ | 70.00 | 55.00 |
| $Al_2O_3$ | 8.50 | 14.00 |
| $Na_2O$ | 2.50 | 3.00 |

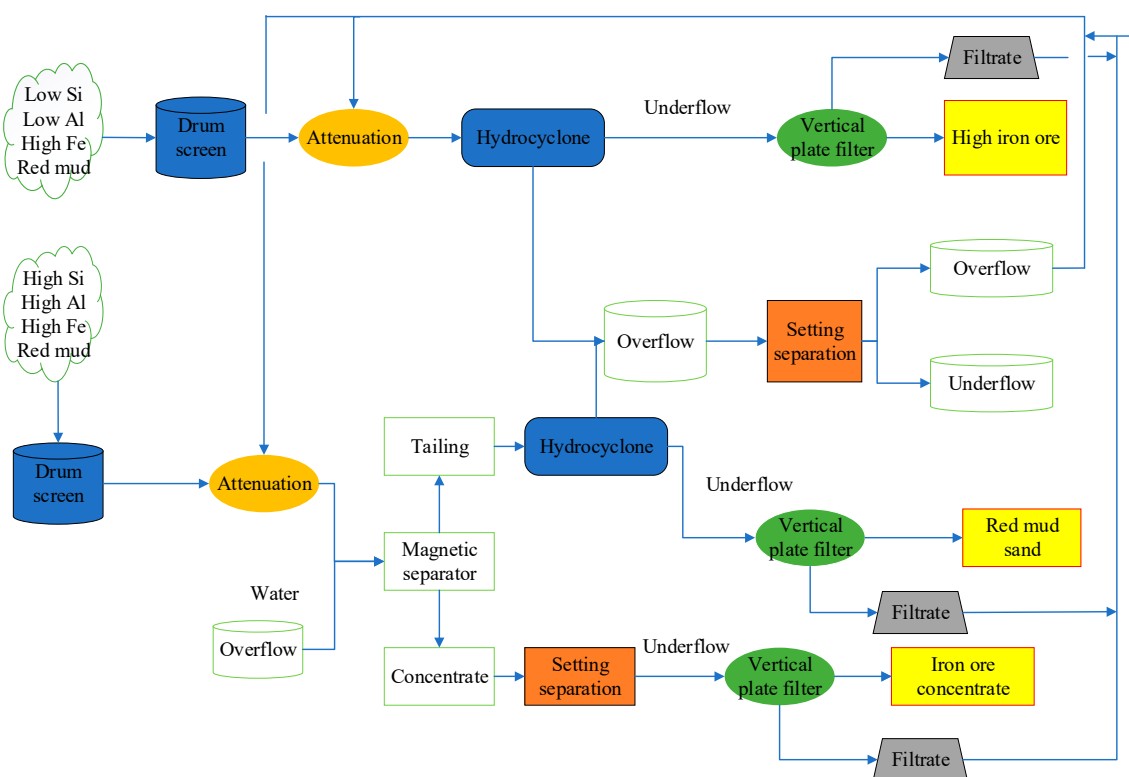

**Figure 11.** Iron separation from Bayer RM used in Shandong Aluminum Company [111].

In addition to extracting Fe from imported bauxite, direct iron extraction from Bayer RM from Pingguo Aluminum Company was also successfully industrialized. In 2009, RM extraction from Bayer RM in Guangxi Aluminum Company was successfully handed over to acceptance (Figure 12). The annual utilization amounts of RM reached 0.2 million tons and the production line was in normal operation. In 2011, Guangxi Aluminum Company constructed a production line of Fe extraction from local RM with an annual handling capacity of 2.2 million tons using the pulsating high-gradient two-stage magnetic separation process. More than 228,800 t of quality iron concentrate can be recovered from the process every year, and the grade of iron concentrate reached more than 55% [112].

Other aluminum plants have borrowed or introduced the corresponding Fe extraction technology and some successfully realized industrialization. Shandong Xinfa Aluminum Company annually recycled 100,000 t of iron oxide from RM since 2009. The RM iron separation project of Guangxi Huayin Aluminum Company was put into operation in 2011. It was estimated that 2.6 million tons of RM will be processed annually, with an annual output of 260,000 t of iron concentrate of 55%. Guangxi Xinfa Aluminum Company adopted pulse high-gradient two-stage magnetic separation technology for the RM iron separation process and produced iron powder with an iron content of nearly 50%, which was put into production in 2013 with an annual digestion of RM of 1.1 million tons and an annual output of about 50% iron powder of 120,000 t. The comprehensive utilization project of 260 t of RM of Yunnan Wenshan Aluminum Company was launched in 2014. The project handled 2.6 million tons of RM annually and produced 55% of the 260,000 t of iron concentrate annually.

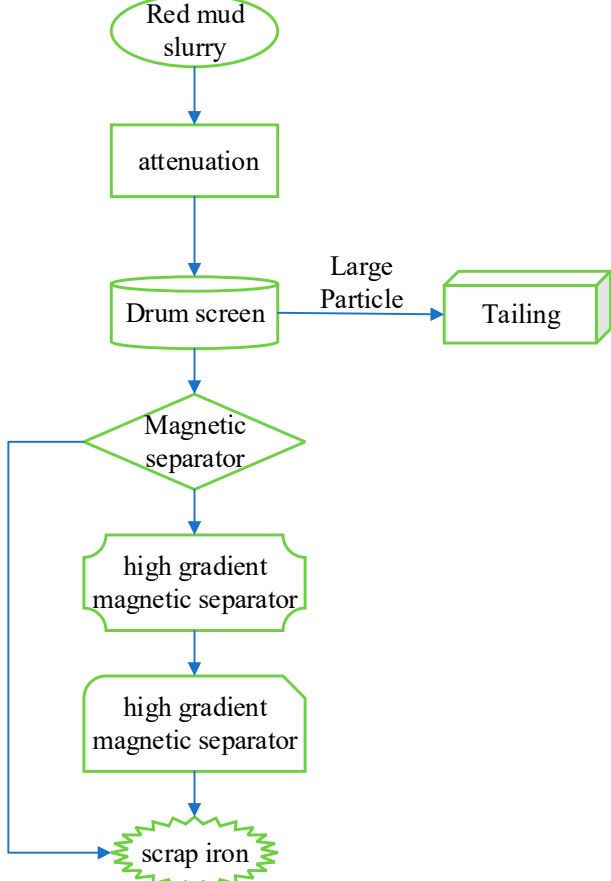

**Figure 12.** Iron extraction from Bayer RM by direct magnetic separation in Pingguo Aluminum Company [42].

### 4.2. Reduction Roasting-Magnetic Separation

Apart some magnetic iron (Fe) compounds, RM also contains many non-magnetic iron compounds. Moreover, the distribution of iron in RM is wide and complex, and uneven particle size and complex compositions of Bayer RM make it difficult for Fe extraction from Bayer RM by direct magnetic separation. Many investigations showed that many iron compounds in RM, such as hematite and goethite, can be converted to magnetite during high-temperature calcination [55,113]. After high-temperature reduction roasting and magnetic separation, Fe-containing compositions in RM could be converted and recycled.

Shandong Aluminum Company conducted reduction roasting using RM as the main raw material and adding some coal-based reducing agent (shown in Figure 13). Roasted products could be converted to iron concentrate after cooling, grinding, dissolving and high-gradient magnetic separating. Then, iron concentrate was reunited with binder by cold solidification, and the lump ore (commonly known as sponge iron) could be obtained [114,115]. The metallization rate, grade and iron recovery rate of the product reached 92.9%, 93.7% and 94.42%, respectively, which could be used as a high-quality iron source for steelmaking, and the residue can be used as raw materials for building materials [116].

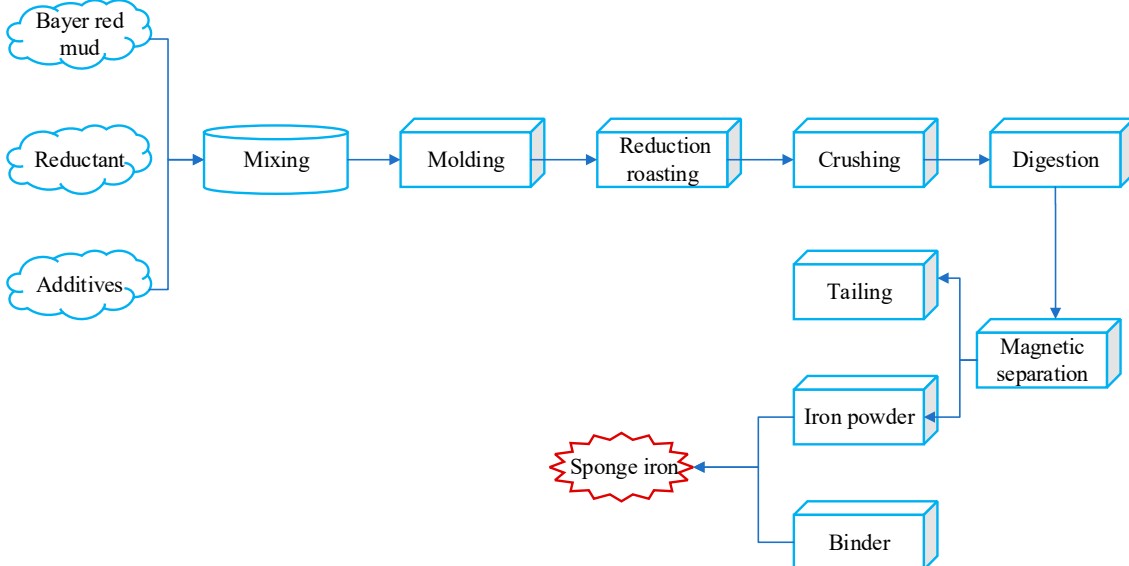

**Figure 13.** Production process of sponge iron from RM [117].

On this basis, Shandong Aluminum Company further improved and treated Bayer RM by direct reduction process using a coal-based bottom furnace, which greatly improved the reduction thermal efficiency of iron, the uniformity of furnace temperature, the metallization rate of finished products, and the design output. RM was extracted by two processes: direct reduction-grinding magnetic separation in a rotary bottom furnace and direct reduction-melting in a gas furnace. The grade and recovery rate of iron in metallic iron powder iron reached 77–85% and 67–78%, respectively. The molten iron contained more than 95% full iron, and the separation effect of slag iron was good [118]. This achievement could efficiently and cheaply deal with most kinds of waste slag and has been put into construction. It was estimated that 300,000 t of RM can be treated annually after the construction completion, and the annual output of 117,000 t of mineral wool and 53,000 t of cast iron blocks would be generated.

Extracting iron from high iron Bayer RM through reduction roasting was also successfully applied in Guangxi Pingguo Aluminum plant. In 2008, the company successfully built a production line of RM magnetic separation iron concentrate production line. The production line could dispose 120,000 t of dry RM annually and the grade of produced iron concentrate was more than 55%, meeting the ore grade requirements of steel smelting enterprises.

The company also cooperated with Guangxi Metallurgical Research Institute, and successfully developed a direct reduction of iron recovery using Bayer RM as a raw material and coal as a reducing agent. Additionally, a production line with an annual capacity of 350,000 t of RM was built, and it produced 83,400 t of sponge iron every year [119].

Actual practices showed that Fe could be effectively extracted through direct magnetic separation and reduction roasting-magnetic separation. On the one hand, the recovery of Fe from RM could create some economic value. One the other hand, it could reduce the amounts of RM emissions and storage, which reduced RM storage costs [120]. Direct magnetic separation has a higher equipment cost but a lower operating cost. Although the Fe compounds extracted from direct magnetic separation can be used as raw materials for steelmaking, the grade and recovery of iron are low. Compared with the direct magnetic separation method, the reduction roasting-magnetic separation method could obtain a higher grade and recovery of iron, but the high-temperature roasting consumes large amounts of energy and release a lot of greenhouse gases, which makes its operation cost high and has a great impact on the environment.

RM contains many valuable metals, such as Fe, Al, Ti, Kh and many kinds of rare-earth elements, which made metal extraction from RM promising. However, metal extraction from RM started late in China, and metal extraction technology lagged behind that of other countries. Fe extraction is the

most widely industrial used metal extraction methods, among which Pingguo Aluminum Company has done the best. Apart from Fe extraction, other valuable metals were just started in China, leaving many key problems still needing to be solved.

## 5. Other Industrial Applications

### 5.1. Calcium Silicon Fertilizer

Calcium silicon fertilizer (a kind of silicon fertilizer) has a good effect on crops due to the fact that it could promote crop growth and enhance crop physiological efficiency and stress resistance. It is widely used in the planting of rice, fruit trees and other crops, and plays a role of resisting pressure and increasing production. Sintering RM contains a high content of calcium silicon and a variety of elements, such as K, P, Cu, Zn, Mn and Mo, which is necessary for the growth of crops. After dehydration at 120–300 °C and activation by grinding to the particle size of 90–150 μm, calcium silicon fertilizer could be prepared [121].

In 2000, Shandong Aluminum Company cooperated with Henan academy of agricultural sciences to develop RM silicon calcium fertilizer, in which the proportion of sintering RM reached 80%. Additionally, soluble silicon and effective calcium was more than 16% and 40%, respectively, and the active ingredients meet the relevant standards (GB/T 1.1-2009) (effective calcium was more than 35%, and soluble silicon was more than 15%). Moreover, silicon, calcium, magnesium, iron and other elements in the product had a high weak acid solubility. The practical utilization of RM calcium silicon fertilizer in silicon-deficient soil showed that this fertilizer had the effect of improving soil structure and crop quality, and increasing production. The ratio of increasing the production ratio of rice, corn, sweet potato, peanut and other crops was generally 8–10% and has formed an annual production scale of 200,000 t [73].

Henan academy of agricultural sciences [122] produced calcium silicon fertilizer using RM as the main raw material and adding a certain amount of silicon fertilizer additives. The calcium silicon fertilizer was successfully used as the fertilizer for peanut cultivation. Practical utilization showed that the large amounts of $SiO_2$ and CaO in RM are beneficial to the growth of peanut, and the yield of peanut was greatly improved by more than 10% after the application of silicon fertilizer. In 2014, the annual output of this kind of RM silicon calcium fertilizer has reached 200,000 t.

RM silicon calcium fertilizer could promote crop growth, enhance crop physiological efficiency and stress resistance, effectively improve crop yield and grain quality, and reduce soil acidity. Additionally, it is suitable for acid, neutral and slightly alkaline soil with little silicon and calcium. The application of RM calcium silicon fertilizer could replenish a large amount of effective silicon and improve soil acidity [121]. However, this technology was rarely used at present, and the main reason was that long-term use was prone to leakage, resulting in groundwater pollution [122].

### 5.2. Flue Gas Desulfurizer

Flue gas desulfurization is more used to control sulfur dioxide ($SO_2$) emissions, which is also one of the most effective measures for clean combustion technology. Generally, low-value limestone or dolomite was selected as sulfur fixing agent for flue gas, but there are some disadvantages, such as low calcium utilization rate and high temperature decomposition of sulfur-fixing products. The mechanism of RM desulfurizer was to use metal ions as catalysts to chemically react with $SO_2$ to form sulfate at a high temperature and use stabilizers to prevent sulfate decomposition [123]. The specific surface area (BET) of RM itself was high and can reach 64–187 $m^2/g$ [58]. Moreover, RM contains many gas-forming phases, such as magnesite and calcite, which could reach higher BET and form porous structure after heat treatment.

In 2009, Shandong aluminum plant successfully prepared a new type of coal desulfurizer from sintering RM, which was successfully applied in its own thermal power plant [83]. Sintering RM extracted from RM yard was pretreated by grinding the particle size to around 125 μm, which made

RM have better sulfur fixation activity. Reasonable mixing with industrial coal powder was carried out by pneumatic conveying equipment according to the ratio of calcium to sulfur of 2.4–3.0, and it was then transported to the boiler synchronously with calcination temperature of 900–1000 °C.

RM not only improved the utilization rate of calcium-based compounds in coal burning, but also reacted with sulfur fixation products to form silicate solid melt at high temperature, which prevented the double decomposition of sulfur fixation products. Moreover, RM had more mesopores and larger specific surface area than limestone after calcination, which improved the chemical reaction speed and depth. In addition, more iron oxide and alkali metal salts in RM also improved the sulfur fixation reaction rate and effective diffusion coefficient [123].

Actual practices showed that RM desulfurizer had higher activity and better sulfur fixation effect compared with traditional desulfurizers such as limestone powder. Additionally, the flue gas desulfurization rate was more than 75%, and the concentration of $SO_2$ in the effluent flue gas was lower than 700 mg/m$^3$, which conformed to the Chinese flue gas emission standard [123]. The production line of RM desulfurizer was successfully built, which produced 300,000 t of RM desulfurizer and consumed 130,000 t of RM per year [124].

This technology could not only digest stored sintering RM but also greatly reduce the cost of RM restoration. The cost of desulfurizer was also reduced due to RM was cheaper than traditional desulfurizer, which could save the raw limestone ore and reduce $CO_2$ emission. It has been estimated that 24 million tons of RM from the front yard could be completely digested in about 20 to 30 years [124].

*5.3. Inorganic Polymer Material*

Polyaluminum ferric chloride (PAFC) is a kind of inorganic polymeric flocculants, which is formed by bridging aluminum chloride and ferric chloride in aqueous solution. PAFC not only has excellent flocculation performance and strong charge neutralization effect like polyaluminum chloride (PAC), but also has strong adsorption and fast precipitation rate like polyferric chloride (PFC). Besides, its chrominance is better than that of PFC, turbidity removal effect and flocculation sedimentation performance are better than that of PAC, which made PAFC an ideal water treatment agent [125].

PAFC can be prepared using aluminum compounds and ferric compounds in RM, which can react with hydrochloric acid and form six coordination molecules such as $[Al(H_2O)_6]^{3+}$ and $[Fe(H_2O)_6]^{3+}$. With the increase in pH value, coordination water in single metal coordination ions was hydrolyzed, and Al and Fe staggered polymerization, which resulted in a higher degree of polymerization of polynuclear hydroxyl aluminum-iron copolymer and formed an inorganic polymer flocculant [126].

Shandong Aluminum Company has overcome a number of technical problems, such as the unstable content of aluminum and iron, the imbalance of salt base degree, the difficult filtration of iron salt colloid and the unstable pH value. In 2016, the project of 50,000 t of composite inorganic polymer water purifier was completed and put into operation. The preparation process was shown in Figure 14 and the chemical composition of RM PAFC was shown in Table 8. Actual practices showed that sewage coagulation turbidity removal rate and COD (chemical oxygen demand) removal rate was more than 95% and 40%, and all data completely reached the standard. This flocculant also showed some advantages of fast reaction speed, good turbidity removal effect, large size and density [126].

This section briefly introduced some novel and feasible RM industrial utilization methods, which showed that RM could be used in an abroad area. Some methods have been forbidden due to the fact that they may generate second environmental pollution, but they provided an ideal for the use of RM, which may be re-activated when limited problems were solved. Some methods have been used in normal for many years, and could be consider whether it could be expanded to other aluminum plants or counties, even all world RM. Some methods were just put into operation, and many subsequent problems were needed to solve.

**Table 8.** Chemical composition of RM polyaluminum ferric chloride (PAFC) [126].

| $Al_2O_3 + Fe_2O_3$ | Basicity | pH | Insoluble Substance |
|---|---|---|---|
| 11.8% | 71.24% | 3.9 | <0.5% |

**Figure 14.** Production process of RM PAFC [126].

## 6. Conclusions

Red mud (RM) is a kind of industrial solid waste generated during the alumina production process. The comprehensive utilization rate of RM is low (only about 4%). The accumulation of RM has caused serious environmental pollution, and the harmless disposal of RM has become a worldwide problem due to its complex physical and chemical composition. Since the establishment of the first aluminum oxide plant, a great deal of research has been done on the utilization of RM and great achievements have been made. Some achievements have also successfully achieved industrialization, providing some ideas for the use of RM.

(1) The use of RM is mainly focused on building materials due to its high slag consumption and low pollution. The engineering properties of RM are poor, which means that RM needs to be pretreated and activated when it is used to produce building materials. Produced building materials have some advantages, such as high early strength, good sulfate and erosion resistance. Normally, industrial wastes are used as activated, which make the maximization of waste control waste become the focus.

(2) Metal extraction from Bayer RM is mainly focused on Fe extraction and can be mainly divided into direct magnetic separation and reduction roasting-magnetic separation. Although both of them have successfully achieved industrialization, there are still some problems limited its broader application. The recovery and grade of Fe by the former is low when the cost of the latter is high, both of which still need more time and researches.

(3)  Apart from building materials and Fe extraction, RM also can be used in other fields, such as calcium silicon fertilizer, desulfurization agent and inorganic polymer material, which provide some new methods for the use of RM and show a broader application prospect.

**Author Contributions:** Conceptualization, W.S. Additionally, L.W.; methodology, L.W.; software, F.L.; validation, L.W., Y.W. Additionally, W.S.; formal analysis, H.Z. (Hua Zeng); investigation, H.Z. (Hua Zeng); resources, H.Z. (Hai Zhang); data curation, H.Z. (Hua Zeng); writing—original draft preparation, H.Z. (Hua Zeng); writing—review and editing, H.Z. (Hua Zeng); visualization, H.Z. (Hua Zeng); supervision, F.L.; project administration, L.W.; funding acquisition, W.S. All authors have read and agreed to the published version of the manuscript.

**Funding:** This work was supported financially by the National Key Research and Development Program of China (No.2018YFC1901901), Natural Science Foundation of Hunan Province (2020JJ5747, 2020JJ5749), National Natural Science Foundation of China (51974365), Young Elite Scientists Sponsorship Program by CAST (Grant No. 2019QNRC001), Innovation-Driven Project of Central South University (No. 2020CX039), Key Laboratory of Hunan Province for Clean and Efficient Utilization of Strategic Calcium-containing Mineral Resources (No. 2018TP1002), Science and Technology Program of Guizhou Province [2017]1092 and Guizhou Mine Division (2020) 25.

**Conflicts of Interest:** The authors declare no conflict of interest.

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
