# Peer review of "Progress on the Industrial Applications of Red Mud with a Focus on China"

_minerals, doi:10.3390/min10090773_

Round 1

Reviewer 1 Report

This review on the industrial application of red mud is not novel, and lack of scientific contribution. The manuscript also has several grammar errors, which makes the reading very difficult and understandable. There are several terms (e.g., 28d, GB 11945-1999) that requires further explanation. The title should read “Progress in the Utilization of Red Mud with a Focus on Its Industrial Applications… in China”? which still is not novel. The manuscript needs to be improved prior to publication. Additional comments/questions:

Abstract

Lines 11-12: “The annual production of RM is large when its average utilization rate is low.” The authors must notice that RM production does not depends on the utilisation rate, but on the aluminium production. Please verify and re-phrase.

Line 13: “… which have caused serious pollution.” The authors must include an example in this statement.

Lines 14-15: What are the authors referring to with “introduced”? Please, re-phrase.

Lines 19-20: “Some industrial utilization methods have not been used due to some problems that cannot be ignored”. The reader does not have a clue about what kind of problem the authors are referring to. Please add examples.

The authors, apparently, have not consider at all the presence of radioactive elements. For instance, thorium that has been found chemically associated to titanium-mineral phases.

Line 31: How much is “huge”? Please give numbers and reference(s). Same line, how much is the consumption of aluminium and steel?

Figure 1: The type of graph is not enough clear. Besides, the legend “Other countries” considers an accumulated amount or average? If so, what’s the deviation? Does the numbers described in Figure 2 correspond to “Other countries”?

Figure 1-2: Do these figures consider aluminium production from ores only? What are the numbers for aluminium production by recycling?

Line 55: “The utilization factor of RM was lower even RM was not used at all in global some areas”. This sentence is not clear. Serious grammar errors, re-phrase.

Lines 59-61: What defines the type of stockpiling? Please, elaborate on this subject.

Line 61-63: Why the water content is important? Please, elaborate.

Line 63-65: Need to be re-phrased.

Line 65: “Also long-termed damming stockpiling of RM would occupy huge land areas”. This sentence is not clear at all! Please, re-phrase.

Lines 69-70: What’s the current cost of RM disposal? Is it still 2% of the Al2O3 price?

Lines 73-74: What about the non-negligible concentration of radioactive elements?

Lines 81-82 serious grammar errors, need to be re-phrased.

Line 84: “…mall number of researches successfully achieved industrialization.” Please give examples and references to sustain this statement.

Figure 4: Why the authors have not included soil remediation, as it has been the case of Alcoa in Australia? Or as a source of CO2 sequestration? Also, as a source for rare-earth elements? Please, elaborate on these subjects.

Lines 90-91 need to be re-phrased.

Line 94-96: What are the authors referring to with “alkaline solubility is good”? Also, there are several grammar errors. Please, verify and correct.

Lines 101-103 needs to be re-phrased, as there are several grammar errors. Besides, what the authors are referring to with “dissolution performance is poor”?

Line 112: Presumably, the authors are referring to the reddish colour of RM? Please, re-phrase.

Line 114: The alkali concentration stated by the authors based on Na2O is not accurate because, in RM, Na is often associated to Al/Si-compounds.

Overall, section 2 lack of data regarding rare-earth element concentration and radioactive elements. These are two important subjects to be included in a review like this one, particularly when the authors are referring to “industrial utilization methods of RM in different regions”. These are fundamental subjects that must be addressed.

Line 150: How much is “a lot of energy”? Please give numbers and references.

Lines 154-156: How Shandong Aluminum Company reduced the alkali content from RM? Please elaborate in this subject.

Line 158: What’s the difference between 425# and 525# Portland cement? Please describe.

Lines 173-174: what the authors mean with “suspended”? was there any legal issues?

Lines 179-181 need to be re-phrased. Please avoid using “And” as noun to start a sentence (this a common grammar issue along the manuscript!).

Lines 185-186: What are the “alkali substances” present in RM? Apparently, the alkali content represents a significant factor to take into consideration to produce cement. However, the manuscript does not contain any information regarding the type of alkaline compounds. In fact, it is necessary to clarify what type of mineral phases defines the alkaline content in RM?

Line 189: “…the traditional sintered brick production process were shown in Fig.6.” It should read “…the traditional sintered brick production process IS shown in Fig.6.”

Lines 188-192: as the authors have given the corresponding description in a past sentence, I presume that in the past bricks were prepared as they described. If so, it would be necessary to state the period (years) when this happened. In addition, it is also important to know how bricks are being produced nowadays.

192: “AND” at the beginning of the sentence, again!

Lines 194-195: What the are the authors referring to with “RM is relatively stable and can be regarded as an inert component when temperature is under 900°C”. RM has some issue to be disposed in the environment mainly due to its high pH, which makes it no inert as the authors stated. Please, re-phrase and fundament.

Lines 199-200: “Long time studies showed that RM brick has good properties and its radioactivity was in an acceptable range”. This sentence confirms that radioactivity is a fundamental subject that need to be addressed in this review. Besides, what are the authors referring to with an acceptable range? How much is acceptable for bricks and/or building materials?

Line 221-223: What patent are the authors referring to? A reference is needed to sustain this statement. How the technology was able to reduce the (unknown) alkali and heavy metals?

Lines 228-229: What are the limits for radioactivity and heavy metal content in building materials? Please elaborate in this subject.

Lines 237-239: Why the authors have given the description on this paragraph as a future sentence?!

Line 240: What the authors are referring to with “global some areas, making producing non-fired bricks”? There is a grammar error in that sentence. Please correct it.

Lines 241-242: The authors have described some mineral phases. However, it is necessary to include the elemental description of each one of these minerals. Also, the authors must verify if the presence of such mineral phases is a common trend in every RM(?), particularly nepheline. What the authors mean with “potential activity”?

Line 243: How the activity index is measured? Why is it an important factor for non-fired bricks?

Line 244: What the authors mean with “exciting agent”? What’s the relation with gel materials? Please clarify.

Lines 247-248: What’s curing?

Line 249: What’s “28d”?

Lines 250-251: What’s the significance of describing pressures? What’s the MU15 or M7.5 brick? And what’s the GB 11945-1999? Please, clarify.

Lines 260-261: What’s a semi-dry pressing moulding process?

Lines 272-274 need to be re-phrase. The text is not enough clear.

Lines 278-279: grammar errors!

Line 383-384: The authors stated that “Practical practices showed that the production line process stability, product quality and economic benefits were good”, but they do not give reference numbers to evaluate the “good” described by the authors.

Lines 400-407: Why is it necessary to melt the mixture of RM, fly ash, carbide slag and coke? It is presumed that part of Fe is removed? If so, what’s the final composition of traces and radioactive elements in the resulting fibre? The authors must be aware that the mere fact of removing Fe from RM, a significant enrichment of the remaining elements can be expected. Please, elaborate on this subject.

Lines 422-428: Why is it necessary to add glass fiber, plant fiber and synthetic fiber?

Line 433-438: What are the “free alkali” present in RM? See my previous question. Also, if RM is “absorbing” the HCl released from the PVC, does part of RM dissolve?

Lines 439-443: What about the content of metal traces and radioactive elements?

Lines 448-450: A reference is needed to sustain this statement.

Section 4 is not novel at all, and the Reviewer does not see any additional contribution to science considering the current state-of-the-art and the wide number of publications in this matter. In fact, why the authors decided to describe only Fe processing, and no other valuable elements?

Line 580: “Calcium silicon fertilizer is a kind of silicon fertilizer, which has a good effect on crops.” What kind of description is this?!

Line 589: What the authors mean with “active ingredients meet the relevant standards”? What are the standards the authors are referring to?

Section 5.2: How sintering RM is produced? And Why?

Line 615: How is it possible that hematite and/or sodium silicoaluminate can form gases? What’s the science behind this statement?

Line 650: What the authors mean with “Al and Fe staggered polymerization”?

Conclusions section

Line 665: What the authors mean with “a kind of industrial solid waste”?.

Regarding metal extraction, the Reviewer does not agree on what Fe is the main focus. Perhaps, only in this manuscript. The authors must re-evaluate their literature survey.  

What about the use of RM on soild remediation, CO2 capture, source for rare-earth elements?

Reviewer 2 Report

This review tried to pile up all the applications or recycling methods for red mud. But rather than list them one by one, I would suggest highlighting the most economical and environmentally friendly applications. The current version is only describing the literature, but what is your analysis? What are your findings? Here are some suggestions to improve the manuscript:
1) The language of the manuscript needs a significant rework. I have a hard time reading your sentences, and the logic is disorder (especially the INTRODUCTION). For example, Line 96, tense & gramma.
2) This review has tons of focus on Chinese literature and utilization in China. But you may also want to look up the research in Australia, Brazi, or North America, and these regions also produce tons of RM. The review misleads me to think China has made the best option for recycling, although the environmental impact is almost unknown for these applications.
3) Figure 1 - This figure is so confusing. Which direction of the y-axis is positive (or negative)?
4) The literature cited by the authors is mostly out of dates, which makes the review depreciated. Also, the details of the literature are not listed in the paper, so I have no idea what testing method or what scenario we are looking at. For example, in Lines 59-70, I suggest emphasizing what types of pollution. For example, heavy metals, high alkaline? or fluoride? There are major issues of the RM. Please check the following literature for more information.
5) Lines 73-74, need a reference here.
6) Lines 82-86, What do you mean by regions? Be specific... region means geological locations or different industries. Also, I am not sure about your objectives for this review. Other than list all the applications of RM, what is your message you want to convey to the audience? What's your suggestion?
7) In general, What are your criteria for "good" or "poor"? Can I trust this word?
8) Lines 210-220, this is a general comment, where are these descriptions come from? It reads like you have translated some news from Chinese to English. Your review should focus on the published data, and give concrete sources of the data.
9) I don't think some of these applications can pass environmental evaluations, and most of them didn't evaluate the leaching at all. Line 294, pH of 7.1 is very neutral for RM, fly ash, and lime. I doubt the research method and data. Why is it neutral? Do you have any clue?
10) Again, these applications may or may not be feasible for different countries. What is your analysis of the applications?

Round 2

Reviewer 1 Report

First of all, it is very difficult to follow the modifications in the new manuscript with respect to the modifications described in the cover letter. I would suggest the authors to submit a new version of the manuscript highlighting the changes, which should be refereed in their reply to Reviewers. While in the cover letter, every change/modification done in the manuscript must be clearly described.  

Regarding point 9 in cover letter: The sentence remains unclear. 

(new) Table 3 contains elements that are not rare-earth nor radioactive elements, such as Ga, V, Zr, Cr, etc. Eventually, the authors may wan to separate them between rare-earth, radioactive and trace elements. 

Regarding point 55 in cover letter: now the another question has arise - why Calcium silicon fertilizer has a good effect on crops?.

Regarding point 58 in cover letter: unless the authors are considering smelting  RM, compounds such as hematite, goethite can release O and/or H2 during their decomposition.

Regarding point 59 in cover letter: the mechanism is not clear yet.

Regarding point 60 in cover letter: The authors must be aware that in EU, RM is not considered as a hazardous substance. If a waste contains more than 1% sodium hydroxide (or any other corrosive chemical that can cause burns) it is indeed classified as a hazardous waste. RM, however, typically contains between 0.2 – 0.6% residual sodium hydroxide, below the 1% threshold. Is this the same case in China? 

Reviewer 2 Report

I am fine with the content. A few works to be considered are: 

Geochemical Characteristics and Toxic Elements in Alumina Refining Wastes and Leachates from Management Facilities

Preparation of red mud-based geopolymer materials from MSWI fly ash and red mud by mechanical activation

These researches showed a good overview of red mud geochemistry or applications. 

Round 3

Reviewer 1 Report

The manuscript has been substantially improved. Although, the manuscript may not have a significant contribution at a fundamental scientific level, it does include some interesting data.